# Use of Space-Time Cube Model and Spatiotemporal Hot Spot Analyses in Fisheries—A Case Study of Tuna Purse Seine

**Ran Xu [1], Xiaoming Yang [1,2,3,4,5]\* and Siquan Tian [1,2,3,4,5]**

1   College of Marine Sciences, Shanghai Ocean University, Shanghai 201306, China;
    m210200712@st.shou.edu.cn (R.X.); sqtian@shou.edu.cn (S.T.)
2   National Engineering Research Center for Oceanic Fisheries, Shanghai Ocean University,
    Shanghai 201306, China
3   Key Laboratory of Sustainable Exploitation of Oceanic Fisheries Resources, Ministry of Education,
    Shanghai Ocean University, Shanghai 201306, China
4   Key Laboratory of Oceanic Fisheries Exploration, Ministry of Agriculture and Rural affairs,
    Shanghai 201306, China
5   Scientific Observing and Experimental Station of Oceanic Fishery Resources, Ministry of Agriculture and
    Rural affairs, Shanghai 201306, China
*   Correspondence: xmyang@shou.edu.cn

**Abstract:** *Katsuwonus pelamis*, or skipjack, is a vital resource in purse seine fishing across the Central and Western Pacific. Identifying skipjack distribution hotspots and coldspots is crucial for effective resource management, but the dynamic nature of fish behavior means these spots are not constant. We used Chinese fishing logbook data from 2010 to 2019 to analyze skipjack resource hotspots and coldspots in a space-time cube. The study revealed 13 spatiotemporal patterns in skipjack Catch per Unit Effort (CPUE). Hotspots (36.53%) were concentrated in the central area, predominantly showing oscillating hotspots (21.25%). The significant effect of the eastern oscillating hotspot continues to be enhanced and extends to the east. Coldspots constituted 63.47% of the distribution, mainly represented by intensifying coldspots (25.07%). The no-pattern-detected type (10.53%) is distributed between coldspots and hotspots. The fishing grounds exhibited longitudinal oscillations of 3°–6° and latitudinal oscillations of 1°–2°. The spatial autocorrelation of cold and hot spot distribution was strong, and the spatiotemporal dynamic changes in skipjack resources were closely related to the El Niño-Southern Oscillation (ENSO) phenomenon. Notably, during 2011–2016, hotspots exhibited an eastward expansion trend, which continued from 2017–2019 due to the influence of fishery management measures, such as the Vessel Day Scheme (VDS) system.

**Keywords:** *Katsuwonus pelamis*; central and western Pacific; purse-seine fishery; space-time cube; emerging hot spot analysis; Mann–Kendall trend test

**Key Contribution:** The space-time cube is seldom used for analyzing fisheries data. This paper introduces this model into fisheries research, providing a solution for maximizing the exploration of fisheries potential and shifting the research focus towards high-yield areas.

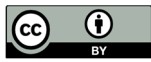

## 1. Introduction

Skipjack (*Katsuwonus pelamis*) is a migratory fish species that is widely found in tropical and subtropical waters. It is an important target species in purse seine fisheries for tuna [1]. The Central and Western Pacific Ocean serves as the largest operational area for purse seine skipjack fishing. In 2021, the catch of tuna in the Central and Western Pacific was 2.034 million tons, accounting for approximately 62% of the global total catch of tuna, and it holds an extremely important position in the world tuna fishery. Currently, skipjack stocks are in good condition with stable replenishment levels, and the catch is below the

Maximum Sustainable Yield (MSY), at a moderately low level of exploitation [2]. However, due to limitations in vessel performance, fishing gear, and captain experience, Chinese fleets still lag behind foreign fleets in terms of net-setting efficiency and single-vessel catch production during fishing operations [3]. Therefore, the accurate identification of the location of cold and hot spots in the distribution of skipjack resources and their spatial and temporal dynamic change trends is of great importance to the exploitation and management of fisheries resources.

Currently, many researchers are actively developing the spatial distribution of cold and hot spots in fishing resources through the utilization of various methodologies, such as model construction and spatial statistics. At present, the Generalized Linear Model (GLMM) is one of the most common methodologies used to calculate the abundance hotspot index of the target species [4]. It is also relatively common to study the spatial and temporal distribution patterns of the target species by constructing Standard Deviation Ellipses (SDEs) [5], or to identify fishing hotspots by studying fleet patterns [6]. Another common approach is to use spatiotemporal statistical models to investigate the activity footprint of the target population and to explore the spatial and temporal clustering and hot spot distribution of unit fishing effort catch [7]. However, existing studies typically only distinguish between cold and hot spots and their distribution, while in reality, the location and status of these spots are not static. Currently, there are only a few studies on identifying trends in the changes of cold and hot spots in trawl fisheries [8,9], and no such studies have been found in other fishery fields.

The ENSO (El Niño-Southern Oscillation) stands as the most potent ocean-atmosphere interaction phenomenon causing global climate variability, with significant implications for worldwide fisheries production [10]. Research revealed that the ENSO phenomenon exerts a pronounced impact on the spatial distribution of purse seine fisheries in the central and western Pacific [11–13]. Some researchers have suggested that changes in skipjack habitat are associated with the zonal displacement of the Equatorial Pacific warm pool. Additionally, variations in the El Niño/Southern Oscillation (ENSO) are also linked to the Intertropical Convergence Zone (ITCZ) and the South Pacific Convergence Zone. In this region, low sea surface salinity results in the formation of a salinity barrier layer, leading to elevated near-surface temperatures that are conducive to the growth of skipjack tuna in the Western Pacific [14]. As a cyclic natural phenomenon, the relationship between ENSO and the dynamic shifts in the distribution of purse-seine fisheries cold and hot spots remains a subject of ongoing discussion and investigation.

The development of the Pacific skipjack fishery in the central and western Pacific Ocean has seen continuous growth since the 1950s. This can be attributed to advancements in fishing gear and methods, increased fishing vessels, and more fishing days [15]. From 2010 to 2013, there was an increase in the number of fishing vessels, which stabilized after 2014. In line with the requirements of international regional fisheries organizations for sustainable fisheries, China has implemented measures to control the scale of its Pacific skipjack fishery since 2016, resulting in stable numbers of fishing vessels and production capacity [16]. Using Geographic Information Systems (GIS) in fishery research can more accurately and reasonably interpret the aggregation characteristics, spatial distribution, and changes in catch trends of fishery resources [17]. By using the space-time cube model and the spatiotemporal hot spot analysis technique in GIS, we can not only find the hot or cold spots of the target attribute values but also determine the intensity and consistency of the hot or cold spots within a certain time unit in order to identify the statistically significant hot and cold spot trends over time. By analyzing the data, the tool can identify new, consecutive, intensifying, persistent, diminishing, sporadic, oscillating, or historical patterns of hot or cold spots in different time intervals. The refined patterns of hotspots or cold spots provide a richer spatiotemporal context beyond single hot or cold spot locations. This aids in uncovering the spatiotemporal evolution patterns of skipjack resources. Currently, this method has been widely applied in research fields such as geography

[18,19], medicine [20], and public transportation [21,22], and introducing this concept provides an effective means for exploring the spatiotemporal information of skipjack resource distribution.

In this study, we have integrated daily fishing log data of all Chinese fishing vessels operating in the Central and Western Pacific Ocean from 2010 to 2019 into daily catch data with a spatial resolution of 0.5° × 0.5°; these data have been used to develop a space-time cube model. Through the application of trend analysis methods, we assess the trends in hot spots and cold spots regarding the spatiotemporal distribution of skipjack resources. Furthermore, our objective is to explore the spatiotemporal evolution patterns of skipjack resources, examining the spatiotemporal dynamics of skipjack fishing hotspots and cold spots in relation to the ENSO phenomenon, fisheries management policies, and other anthropogenic influences. This research aims to provide valuable insights into the development of Central and Western Pacific skipjack resources and the formulation of effective fishery management policies.

## 2. Materials and Methods

### 2.1. Data Sources

The catch data for skipjack in the Central and Western Pacific Oceans were extracted from the fishing logbooks of all Chinese-flagged vessels operating between 2010 and 2019 (Table 1). The data were selected based on the following criteria: the target catch was skipjack, the fishing method was purse seine fishing, and the fishing operations were conducted in the Central and Western Pacific Ocean (138° E–148° W, 15° N–15° S, Figure 1). The recorded information encompassed the year, month, day, longitude, latitude, fishing days, and catch quantity measured in tail numbers, among other relevant variables.

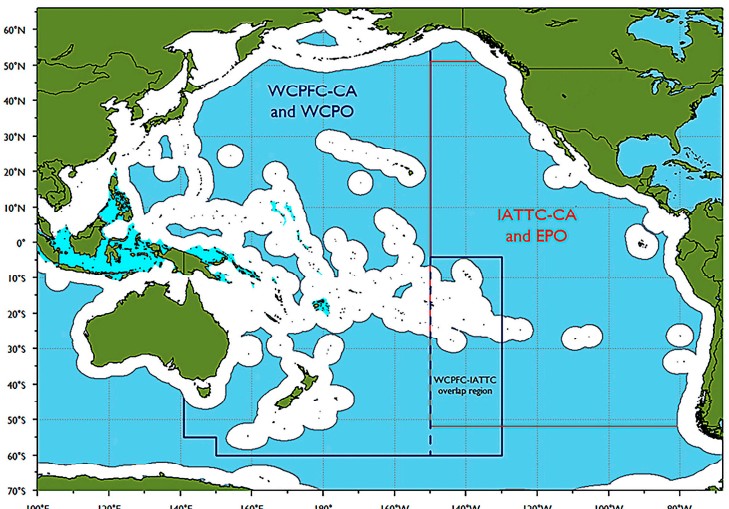

**Figure 1.** Important national, regional and management zones in the Pacific. The WCPFC Convention Area (WCPFC-CA) is outlined in dark blue, the IATTC Convention Area (IATTC-CA) area is outlined in red. The western and central Pacific Ocean (WCPO) includes all of the WCPFC-CA, minus the overlap with the IATTC-CA; the eastern Pacific Ocean (EPO) is coincident with the IATTC-CA. Pacific nation EEZs are outlined in grey and archipelagic waters are shaded turquoise [23].

**Table 1.** The information on the fishing days for the Central and Western Pacific skipjack tuna purse seine fishery operations. The term "fishing day" represents "Days fishing and searching (effort)".

| Year | Total Vessels (pcs) | Total Fishing Days (d) | Catch (t) |
|---|---|---|---|
| 2010 | 12 | 2222 | 37,705 |
| 2011 | 14 | 2353 | 56,357 |

| 2012 | 15 | 2546 | 66,237 |
| 2013 | 21 | 4883 | 115,351 |
| 2014 | 25 | 4301 | 107,529 |
| 2015 | 24 | 2788 | 64,045 |
| 2016 | 26 | 4567 | 121,726 |
| 2017 | 24 | 5386 | 124,677 |
| 2018 | 23 | 5016 | 168,410 |
| 2019 | 15 | 3575 | 132,913 |

*2.2. Data Preprocessing*

We initiated our study with data preparation and preprocessing steps, where we adjusted the spatial resolution of the target dataset to 0.5° × 0.5°. The data had a daily time resolution, and we integrated the catch production and fishing effort data on a daily basis within the 0.5° × 0.5° spatial cells. The indicator chosen to represent the status of the skipjack fishery resources is the Catch per Unit Effort (CPUE), with the following calculation formula:

$$CPUE = \frac{U_{catch}}{f_{days}} \tag{1}$$

In the equation: '$U_{catch}$' represents the cumulative catch within a spatial cell of 0.5° × 0.5°, measured in metric tons (t), '$f_{days}$' represents the cumulative operational duration within a unit, measured in days (d), with a time scale of one day.

*2.3. Data Analysis Methods*

2.3.1. Spatial Autocorrelation Analysis

The establishment of scientific spatial analysis or forecasting models requires not only attention to the correlation of data but also the study of their two-dimensional spatial relationships. To solve this problem, methods such as selecting statistical indicators, adaptive modeling, and semi-empirical models are usually used [24]. Spatial autocorrelation analysis is a branch of spatial statistics, and the Moran's I and Local Moran's I statistics under this model have been widely used to study two-dimensional relationships in data [25–27].

Moran's I is a statistical measure that enables the simultaneous analysis of spatial autocorrelation among both the attributes and locations of features. The calculation formula for Moran's I is as follows:

$$\frac{\sum_{i=1}^{n}\sum_{j=1}^{m} w_{ij} (x_i - x_m)(x_j - x_m) / \sum_{i=1}^{n}\sum_{j=1}^{m} w_{ij}}{\sum_{i=1}^{n} (x_i - x_m)^2 / n} \tag{2}$$

where: $x_i$, $x_j$ represent the attribute values of units *i* and position *j*, respectively.

$$x_m = \frac{1}{n}\sum_{i=1}^{n} x_i \tag{3}$$

$W_{ij}$ represents the spatial weight matrix, which defines the neighborhood relationships among spatial objects. Moran's I takes on positive values when the attribute values of neighboring units are similar, negative values when they are dissimilar and tends towards zero when attribute values appear randomly. The value of Moran's I ranges between −1 (indicating dispersed patterns) and +1 (indicating clustered patterns), with a value around 0 indicating a random pattern.

2.3.2. The Space-Time Cube Model

The space-time cube model is a spatiotemporal data model within GIS that integrates spatial, temporal, and attribute information of geographic phenomena. It enables the reconstruction of historical states, tracking of spatiotemporal changes, and prediction of development trends of data [28,29]. This model facilitates 3D visualization and data analysis. By aggregating all sample points into spatiotemporal columns, the data is organized into a structured format using NetCDF (Network Common Data Form) as the spatiotemporal data structure. Within each column, calculations are performed on the points, and specified attributes are aggregated to calculate statistical data for all summary fields. The Mann-Kendall trend analysis is employed to assess the trend of column values across time at each location, and compares the trend data bin values created at each location with the previous unit value, with positive values indicating a higher presence, negative values indicating a decrease, and zero values indicating no change. This analysis yields the time series trend for the entire study area.

Conceptually, this data structure can be visualized as a three-dimensional cube composed of spatiotemporal columns. The x and y dimensions represent the spatial locations of geographic entities, while the z dimension represents time. Each column has a fixed position in space (x, y) and time (z), and columns covering the same (x, y) area share the same location ID. Columns with the same duration share the same time step ID. Columns associated with the same physical location share the same position ID and can be combined to represent time series, while columns sharing the same time step interval can be combined to form time slices [30].

2.3.3. Spatiotemporal Hotspot Analysis

The Getis-Ord Gi* statistic is a novel tool used for spatiotemporal hotspot analysis of data in the space-time cube [31–33]. It allows for the identification of spatiotemporal trends, as well as hotspots or coldspots of data on a spatiotemporal scale (Table 2). This method retrieves and analyzes adjacent columns in both time and space, calculating Z scores, *p* values, and other information associated with each column based on appropriate neighborhood distance and neighborhood time step parameters [34]. By examining significant clusters of high or low values within a region, hotspots or coldspots of the target attribute values can be identified, and the strength and consistency of hotspots or coldspots within a certain time step can be determined. The spatiotemporal hotspot analysis tool also calculates the correlation between columns. In this study, the global Moran's index tool is employed to calculate spatial autocorrelation, assisting in the determination of suitable neighborhood distances and neighborhood time steps. Additionally, the hotspot analysis enables two-dimensional or three-dimensional visualization of the space-time cube, facilitating the visual identification of spatiotemporal data patterns.

**Table 2.** Trend significance classification categories.

| Trend Bin | Z-Score | *p*-Value | Trend | Remarks |
|---|---|---|---|---|
| −3 | <−2.58 | 99% | Decline with 99% confidence level | Cold spot with 99% confidence level |
| −2 | −2.58 ~ −1.96 | 95% | Decline with 95% confidence level | Cold spot with 95% confidence level |
| −1 | −1.96 ~ −1.65 | 90% | Decline with 90% confidence level | Cold spot with 90% confidence level |
| 0 | −1.65 ~ 1.65 | — | Non-significant trend | Non-statistically significant hot or cold spots |
| 1 | 1.65 ~ 1.96 | 90% | Up with 90% confidence level | Hot spot with 90% confidence level |

| 2 | 1.96 ~ 2.58 | 95% | Up with 95% confidence level | Hot spot with 95% confidence level |
| 3 | >2.58 | 99% | Up with 99% confidence level | Hot spot with 99% confidence level |

According to the Z scores and *p* values of each column, the patterns of columns in the spatiotemporal hotspot analysis can be classified into different categories as summarized in Table 3.

**Table 3.** Classification of Hotspot and Coldspot Trends. The classification categories describe the patterns of hotspots and coldspots observed in the spatiotemporal analysis [34].

| Pattern Name | Definition |
|---|---|
| No Pattern Detected | Does not fall into any of the hot or cold spot patterns defined below. |
| New Cold/Hot Spot | A location that is a statistically significant cold/hot spot for the final time step and has never been a statistically significant cold/hot spot before. |
| Consecutive Cold/Hot Spot | A location with a single uninterrupted run of statistically significant cold/hot spot bins in the final time-step intervals. The location has never been a statistically significant cold/hot spot prior to the final cold/hot spot run and less than ninety percent of all bins are statistically significant cold/hot spots. |
| Intensifying Cold/Hot Spot | A location that has been a statistically significant cold/hot spot for ninety percent of the time-step intervals, including the final time step. In addition, the intensity of clustering of high counts in each time step is increasing overall and that increase is statistically significant. |
| Persistent Cold/Hot Spot | A location that has been a statistically significant cold/hot spot for ninety percent of the time-step intervals with no discernible trend indicating an increase or decrease in the intensity of clustering over time. |
| Diminishing Cold/Hot Spot | A location that has been a statistically significant cold/hot spot for ninety percent of the time-step intervals, including the final time step. In addition, the intensity of clustering in each time step is decreasing overall and that decrease is statistically significant. |
| Sporadic Cold/Hot Spot | A location that is an on-again then off-again cold/hot spot. Less than ninety percent of the time-step intervals have been statistically significant cold/hot spots and none of the time-step intervals have been statistically significant hot/cold spots. |
| Oscillating Cold/Hot Spot | A statistically significant cold/hot spot for the final time-step interval that has a history of also being a statistically significant hot/cold spot during a prior time step. Less than ninety percent of the time-step intervals have been statistically significant cold/hot spots. |

## 3. Results

### 3.1. Spatial Autocorrelation of the CPUE of Skipjack

The spatial autocorrelation analysis was conducted to examine the spatial pattern of the CPUE of skipjack in the study area. The results of the Global Moran's I statistical test indicated that the CPUE of skipjack exhibited a clustered pattern, meaning that areas with

high CPUE values were spatially associated with neighboring areas with high CPUE values, and the same applied to areas with low CPUE values. To determine the optimal neighborhood distance for constructing the space-time cube, incremental spatial autocorrelation analysis was performed using multiple distance bands. The Global Moran's I was computed for each distance band, providing a distance threshold and a corresponding Z-score. The distance threshold represents the spatial scale at which the most significant clustering occurred for each year, and it serves as the optimal spatial scale for conducting spatial analysis in that specific year. The results revealed that the optimal neighborhood distance varied for each year from 2010 to 2019, ranging from 22,799 m to 67,242 m. However, when considering the entire decade, the overall optimal spatial scale was determined to be 55,500 m, which captures more spatial information and retains more research details at this scale. This distance was selected as the optimal neighborhood distance for constructing the space-time cube.

Furthermore, the trend analysis of the Z-score indicated an upward trend over time, suggesting that the clustering of skipjack CPUE became more pronounced as the years progressed. These findings provide valuable insights into the spatial autocorrelation of skipjack CPUE, highlighting the presence of significant clustering patterns and indicating the optimal spatial scale for conducting further spatial analysis.

*3.2. Space-Time Cube Model of Skipjack*

The analysis data consisted of a theoretical sampling of 3652 times, and the actual sampling included 3588 instances, comprising a total of 28,490 CPUE data points. Based on the spatial autocorrelation analysis, the optimal spatial scale for clustering was determined to be 55,500 m. Using the logbook data of the Pacific skipjack tuna purse seine fishery in the Central and Western Pacific from 2010 to 2019, a space-time cube model was constructed. The model had a time step interval of 1 year and a neighborhood distance of 55,500 m.

The space-time cube aggregated the 28,490 CPUE data points into 5310 0.5° × 0.5° spatial cells over 10 time step intervals. Each 0.5° × 0.5° spatial cell represented a square area of 55,500 m by 55,500 m. The entire space-time cube covered an area of 654,900 m from west to east and 2,497,500 m from north to south. The time period covered by the space-time cube spanned 10 years. Out of the total 5310 locations, 2270 locations (42.75%) contained at least one data point for at least one time step interval. These 2270 locations comprised 22,700 space-time bins, of which 8076 (35.58%) had point counts greater than zero. The trend analysis indicated a statistically significant increase in CPUE point counts over time, suggesting a rise in the CPUE of skipjack during the study period.

The space-time cube model allowed for two-dimensional and three-dimensional visualizations, which facilitated the identification of spatiotemporal patterns. The 2D visualization provided an overview of the overall trend throughout the study period, while the 3D visualization displayed the historical state and changes of each fishing ground represented by each 0.5° × 0.5° spatial cell over time. Figure 2 depicts the 3D representation of the space-time cube, illustrating the total catch (in tons) at each location and year.

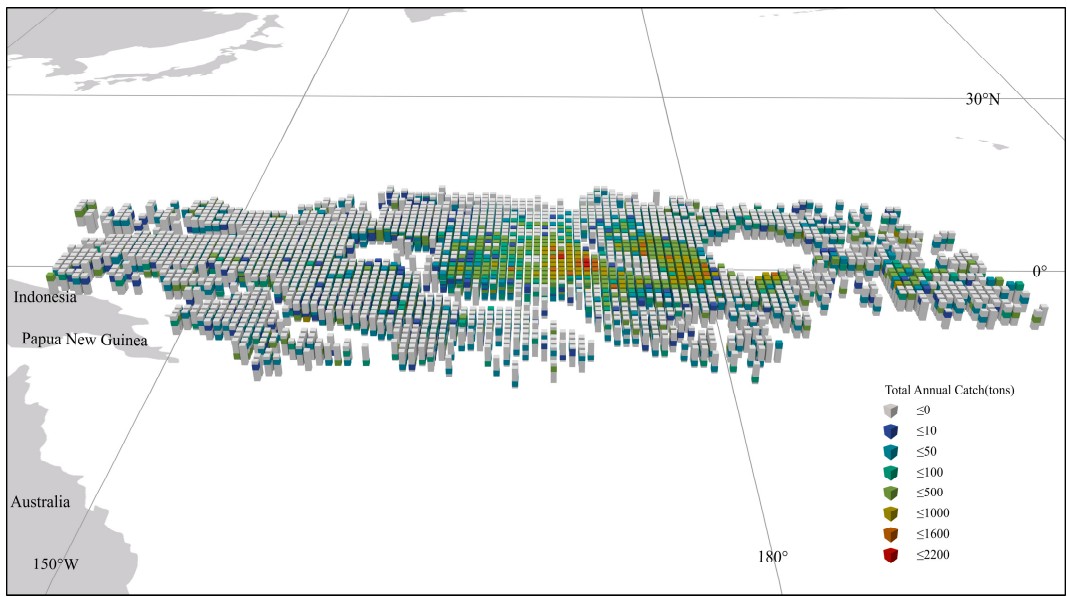

**Figure 2.** Spatio-temporal cube displayed in three-dimensional mode. Layers represent skipjack catch data by year.

### 3.3. Mann–Kendall Trend Test for CPUE of Skipjack

The Mann–Kendall trend test was conducted using the space-time cube model to analyze the CPUE of skipjack over the entire study period. The results (Figure 3) indicate that there is no significant overall trend observed in the CPUE of skipjack. However, there are localized trends of increase or decrease. In the central area of the study region, a total of 568 spatial cells of 0.5° × 0.5° exhibit a significant upward trend in CPUE, suggesting the presence of potential fishing hotspots. These locations are primarily concentrated in the geographical range of 163° E–179.5° W and 5.5° N–6° S. Moving outward from this central area, there are 708 fishing net locations with insignificant trends, which could be attributed to fluctuations in the fishery. On the periphery of the study area, there are 994 locations showing a decreasing trend in CPUE over time. These areas may represent fishing cold spots, where the CPUE of skipjack has declined. The Mann–Kendall trend test provides valuable insights into the overall changes in CPUE and identifies both significant and insignificant trends across the study area. These findings can be utilized to inform fisheries management strategies and target specific regions for conservation or intervention efforts.

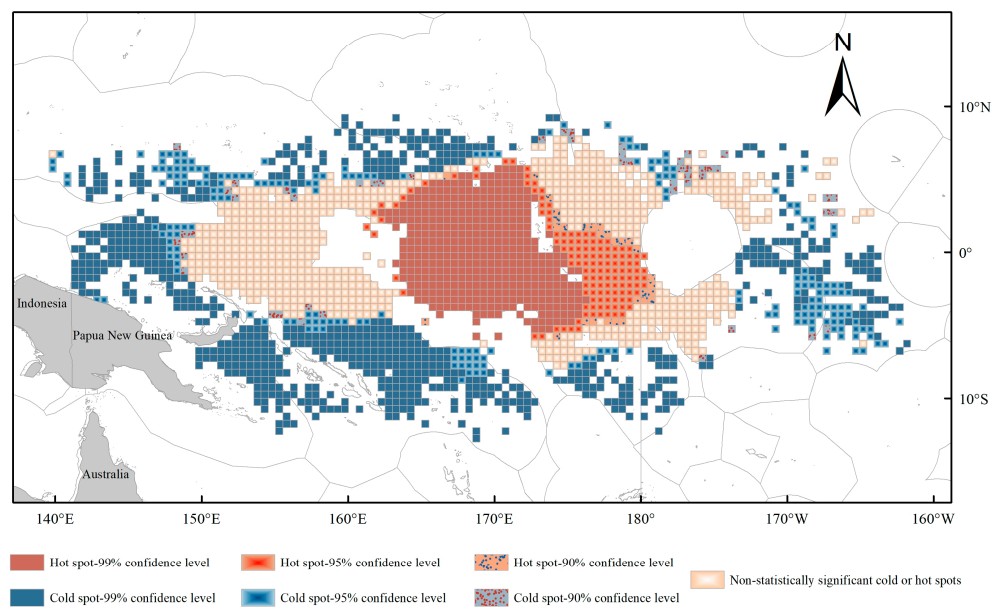

**Figure 3.** Mann–Kendall trend test results for CPUE of skipjack, with cool and warm colors representing areas where cold and hot spots are likely to occur, respectively.

### 3.4. Spatiotemporal Distribution of Hot and Cold Spots for CPUE of Skipjack

The spatiotemporal hotspot analysis using the space-time cube model revealed the distribution of hot and cold spots for CPUE at each fishing net location in the study area. Figure 4 visualizes the results, where warm colors represent hot spots and cold colors represent cold spots. Out of the total 2270 fishing net locations analyzed, 2031 locations (89.42%) exhibited hot or cold spot trends. These locations showed varying degrees of clustering in CPUE over time. Table 4 provides an overview of the thirteen results obtained from the spatiotemporal hotspot analysis. Among these results, a total of 742 fishing net locations were identified as fishing hotspots, accounting for 32.69% of the total locations. These hotspots were mainly concentrated in the central area of the study region, spanning from longitude 151° E to 178° W and latitude 5° N to 6° S. These regions experienced significant clustering of high CPUE values over time. The fishing net locations surrounding the hotspots showed no pattern detected, with a total of 239 locations (10.53%) falling into this category. These locations did not exhibit significant clustering of CPUE values. Moving outward from the hotspots and surrounding areas, a total of 1289 fishing net locations (56.78%) displayed a cold spot pattern. These locations experienced a decreasing trend in CPUE over time.

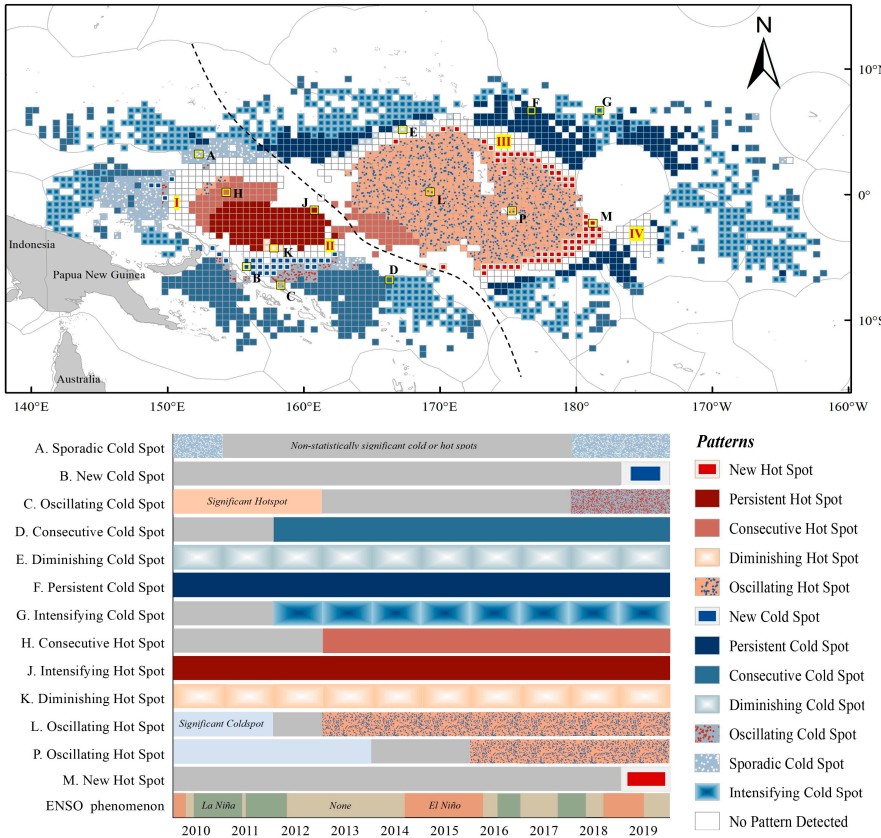

**Figure 4.** Spatiotemporal distribution of hot (warm colors) and cold (cold colors) of skipjack tuna CPUE, showing 13 spatial and temporal distribution patterns, with stable distribution patterns of cold and hot spots.

The spatiotemporal distribution of hot and cold spots provides valuable insights into the variability and clustering patterns of CPUE in different areas of the study region. This information can aid in identifying key fishing hotspots, understanding spatial dynamics, and informing targeted fisheries management strategies to maximize catch efficiency and conservation efforts. Based on Figure 4 and Table 4, the following observations can be made:

(1) There are no intensifying, sporadic, historical hotspots, or historical cold spots in the study area, indicating that the spatial position of the hotspot area is constantly fluctuating. The development and utilization of skipjack resources in the study area is relatively mature, with stable production, and has been developed for a long time. Therefore, there has not been an area with continuously increasing clustering strength in 2010–2019.

(2) In the study area, there were 89 consecutive hotspots and 106 persistent hotspots between longitude 151.5° E–168.5° E and latitude 1.5° N–5° S (Figure 4). This result indicates that there are stable fishing grounds in the study area, which are constantly changing due to phenomena such as El Niño and La Niña, but there are still continuous and stable fishing grounds among them. Four diminishing hotspots were detected, which were produced around persistent hotspots. They showed hotspots 90% of the time but their clustering strength decreased overall, and this decrease was statistically significant.

(3) The new hotspots in the study area (63 in total) are concentrated between 166.5° E and 178° W, and between 5.5° N and 6.5° S. These locations exhibited statistically

significant hotspots in 2018–2019 (Table 4), but not in previous years. The new hotspots are generated around oscillating hotspots (Figure 4).

(4) In the study area, the proportion of intensifying cold spots is the highest, with 559 detected in total, 307 consecutive cold spots, 242 persistent cold spots, and 110 sporadic cold spots. There are also 33 new cold spots, 29 oscillating cold spots, and 9 diminishing cold spots (Table 4). The results show that these cold spots are mainly located in the boundary area of the study area (Figure 4), indicating that the fishing operation mode in China is gradually maturing, and the detection of fishing grounds tends to be stable.

(5) The fishing net positions around various hotspots in the study area show no pattern detected and are distributed in 239 locations (Table 4). The distribution range is between 150° E and 174° W, and between 6.5° N and 7.5° S, spreading out from the hotspot area and forming a shape that is almost circular. The distance between the fishing net positions and the nearest hotspots is approximately 0.5°–1.5° (Figure 4). There is no unified pattern in this area, and it does not belong to any established cold or hot spot pattern. Dividing at 162.5° E, the area to the west of 162.5° E shows an irregular alternating pattern of cold and hot spots, with occasional hot spots appearing irregularly. The area to the east of 162.5° E shows an irregular pattern of cold spots, with statistical significance fluctuating.

**Table 4.** Results of spatiotemporal cube detection of the distribution of skipjack in the Central and Western Pacific Ocean.

| Type | Number | Percentage | Locations | Main Periods of Occurrence |
|---|---|---|---|---|
| New Hot Spot | 63 | 2.78% | 166.5° E–178° W, 5.5° N–6.5° S | During 2019 |
| Consecutive Hot Spot | 89 | 3.92% | 151.5° E–158° E, 1.5° N–3° S 162° E–168.5° E, 0.5° S–5° S | 2010–2011: no salient features; 2011–2012: partially significant hotspots; 2013–2019: Significant hotspots. |
| Intensifying Hot Spot | 0 | 0.00% | None | None |
| Persistent Hot Spot | 106 | 4.67% | 153° E–162.5° E, 0.5° S–4.5° S | 2010–2019: Significant hotspots |
| Diminishing Hot Spot | 4 | 0.18% | 157° E–158.5° E, 3.5° S–4.5° S | 2010–2019: Significant hotspots; During 2019: Hotspot clustering intensity decreases. |
| Sporadic Hot Spot | 0 | 0.00% | None | None |
| Oscillating Hot Spot | 480 | 21.15% | 152° E–154.5° E, 1.5°N–0° | During 2010: Significant cold spots; During 2011: No significant features; During 2012: some areas are significant hotspots; 2013–2019: Significant hotspots. |
| | | | 163° E–179.5° W, 5.5° N–6° S | Between 163.5° E–180° E, 5° N–5° S — West of 174° E: 2010–2011: vast majority of significant cold spots; 2011–2012: some regions without significant features; 2011–2012: some regions without significant features; 2012–2014: some regions are significant hotspots; 2014–2019: overwhelmingly significant hotspots. East of 174° E: 2010–2015: overwhelmingly significant cold spots; 2015–2018: some regions are significant hotspots; 2018–2019: Significant hotspots. |

| Type | Count | Percentage | Coordinates | Region | Description |
|---|---|---|---|---|---|
| | | | | Marginal areas | 2010–2017: most regions show a process of change from significant cold spots to no significant features; 2017–2018: some regions are significant hotspots; 2018–2019: Significant hotspots. |
| Historical Hot Spot | 0 | 0.00% | None | | None |
| New Cold Spot | 33 | 1.45% | 153° E–162.5° E, 4° S–6.5° S | | During 2019 |
| Consecutive Cold Spot | 307 | 13.52% | 139.5° E–149.5° E, 1° N–7° N 149.5° E–166.5° E, 5° S–12.5° S | | 2010–2012: no distinguishing features; 2012–2015: some areas are significant cold spots; 2015–2019: Significant cold spots. |
| Intensifying Cold Spot | 559 | 24.63% | 141° E–151° E, 2° N–5° S 166° E–175° W, 6.5° S–12° S 143° E–170° W, 3.5° N–9° N 174° W–178° W, 3.5° N–6.5° S | | 2010–2011: small proportion with no significant features; 2011–2019: cold spots of significance and gradually increasing clustering. |
| Persistent Cold Spot | 242 | 10.66% | 163.5° E–171° W, 1.5° N–8° N 173.5° E–175.5° W, 3.5° S–8° S | | 2010–2019: Significant cold spots |
| Diminishing Cold Spot | 9 | 0.40% | 162° E–171° E, 3° N–6.5° N | | 2010–2019: Significant cold spots; 2018–2019: weakening intensity of clustering. |
| Sporadic Cold Spot | 110 | 4.85% | 144.5° E–158° E, 4.5° N–3.5° S | | During 2010: mostly significant cold spots; 2011–2017: partially non-significant; 2017–2019: Significant cold spots. |
| Oscillating Cold Spot | 29 | 1.28% | 153.5° E–162.5° E, 5° S–7.5° S | East of 156° E | During 2010: No distinguishing features; 2011–2013: Significant hotspots; 2013–2017: gradual change from salient hotspot to no salient features; 2017–2019: Significant cold spots. |
| | | | | West of 156° E | 2010–2013: Prominence hotspots; 2013–2017: partially unremarkable; 2017–2019: Significant cold spots. |
| Historical Cold Spot | 0 | 0.00% | None | | None |
| No Pattern Detected | 239 | 10.53% | 150° E–174° W, 6.5° N–7.5° S | | No apparent pattern |

(6) According to the information provided(Figure 4), there are 480 oscillating hotspots in the study area. These hotspots are concentrated between 163° E and 179.5° W, and between 5.5° N and 6° S. Additionally, there are a smaller number of hotspots appearing between 152° E and 154.5° E. The variation process of oscillating hotspots is complex. Figure 5 illustrates the number of years that each location in the area has remained as a hotspot. Locations adjacent to consecutive hotspot areas have consistently shown significant hotspots from 2012 to 2019. Positions that expand northeastward from this area have shown significant hotspots from either 2013 to 2019 or 2014 to 2019. This forms an approximately fan-shaped area between 163.5° E and 174° E and between 4° N and 5° S. In terms of longitude, as the fishing positions expand eastward, locations east of 174° E show significant hotspots from either 2015 to 2019 or 2016 to 2019. The fishing positions at the north, east, and south boundaries of the

area showed significant hotspots only in 2018–2019. Overall, from 2010 to 2019, most of the central area positions directly transformed from significant cold spots to significant hotspots. However, at most of the boundary positions, the change process involved significant cold spots, followed by no significant pattern, and then significant hotspots. In this study, we believe that the occurrence of oscillating hotspots indicates that the frequency and amount of fishing catches in this area are irregular and mainly influenced by environmental changes.

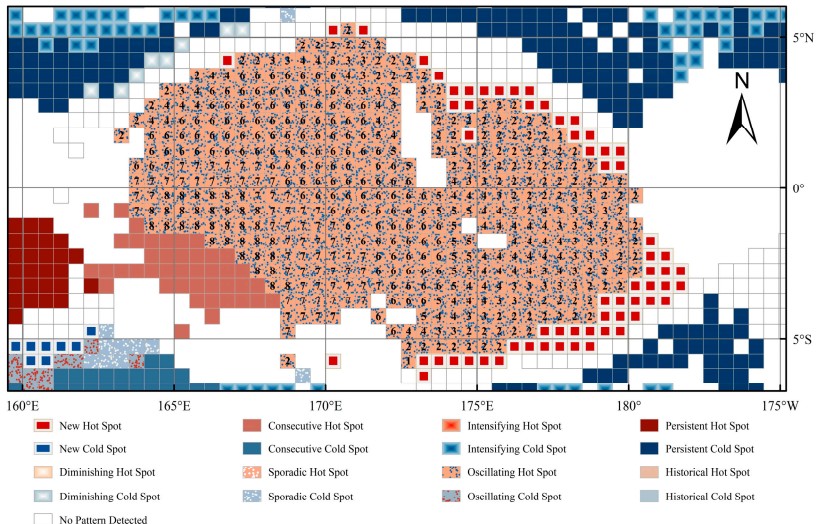

**Figure 5.** The change in the number of years of sustained significant hotspots in oscillating hotspot regions. The numbers in the figure indicate the duration of significant hotspots manifested at the location.

Figure 6 is a layered space-time cube showing the variability of cold and hot spots in CPUE of purse seine fishery in the Central and Western Pacific in both space and time. Each layer represents the hotspot analysis results of CPUE for that year. According to the results, from 2010 to 2013, the fishing hotspots in the study area were primarily concentrated in the western waters between 151° E and 163° E. However, after 2013, another center of fishing hotspots emerged in the waters between 163° E and 175° E. Over time, these hotspots expanded eastward and reached as far as 178° W in 2019. The most persistent hot spot area over the past decade has been concentrated in the western waters of the study area, which has shown a slight trend of contraction and expansion in different years, leading to the appearance of some oscillating hot spots and diminishing hot spots (especially between 2011 and 2013).

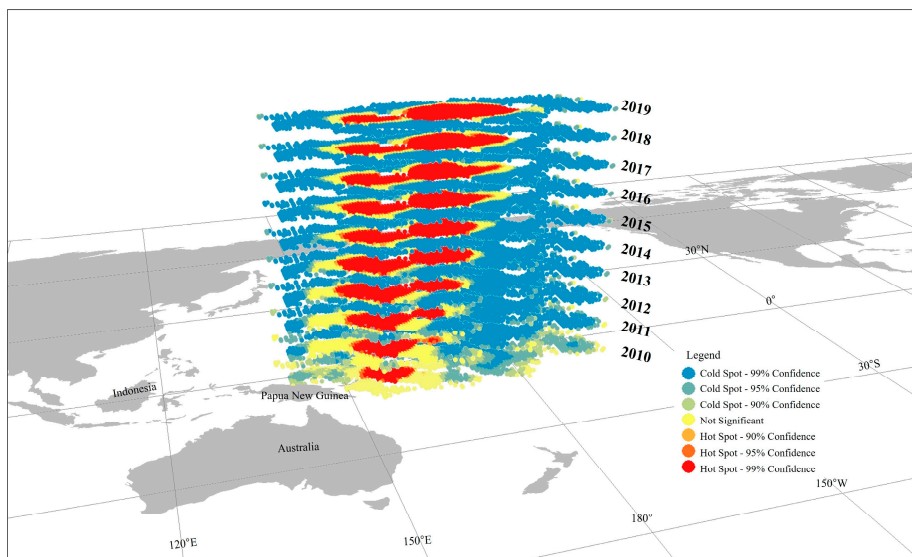

**Figure 6.** Results of the three-dimensional hotspot analysis of total annual catch, with each layer representing a year, analyzing the hotspot movement trend of skipjack CPUE from year to year, with an overall eastward trend between years.

## 4. Discussion

### 4.1. Relationship between ENSO Phenomenon and Spatiotemporal Patterns of the Western and Central Pacific Skipjack CPUE

This study investigates the temporal and spatial dynamics of the CPUE of Western and Central Pacific skipjack using the space-time cube model and emerging hotspot analysis from 2010 to 2019 during Chinese fishing vessel operations. The study identified 13 spatiotemporal distribution patterns through hotspot analysis, including 32.69% hot spot patterns, 56.78% cold spot patterns, and 10.53% no pattern detected. The main hot spot patterns include oscillating hot spots, persistent hot spots, and consecutive hot spots, which are concentrated in the central area of the research region. The primary cold spot patterns include intensifying cold spots, consecutive cold spots, and persistent cold spots, which are stably distributed around the study area.

The results show that the distribution of cold and hot spots of skipjack CPUE in the study area has a strong regularity, and at the same time, a complex pattern of left-right oscillation occurs in the middle of the stable distribution of cold and hot spots over a long time span (see Figure 4, regions I-IV). The survival activities of skipjack are closely related to climate changes [35–37], and we hypothesize that this oscillatory phenomenon is primarily influenced by the ENSO phenomenon. From the perspective of the formation patterns of fishing grounds, it is evident that the environmental conditions within small-scale marine areas exhibit continuity. Drawing upon established empirical knowledge, such as Tobler's First Law [38], it can be inferred that under continuous and similar environmental backgrounds, the occurrence of continuous fishing grounds is highly probable. However, within the findings of this study, the "No Pattern Detect" phenomenon emerges around the stable fishing grounds area (see Figure 4, regions I-IV). The "No Pattern Detect" represents the alternating appearance of fishing cold and hotspots in the given time intervals within the region. The frequency is not fixed, and there is no consistent regularity. The location of the "No Pattern Detect" phenomenon and the amplitude of the oscillations are highly consistent with the left-right movement of warm pools in the western Pacific Ocean during ENSO events. This study highlights the regularity observed in the distribution of these spots, indicating a close relationship between interannual scale climate changes and the dynamic shifts in catch.

Skipjack is a cluster fish species that exhibits strong spatial autocorrelation in its resource distribution at a macro level and has localized features of uneven "cold" and "hot"

distribution [39]. The study finds that the CPUE of Western and Central Pacific skipjack is not uniform in time and space(Figure 4), with the Mann–Kendall trend test and emerging hotspot analysis results identifying a common hotspot core area, the results show that from 2010 to 2019, the CPUE hotspots of Western and Central Pacific skipjack were concentrated in the central area of the Western and Central Pacific, and the center of the annual CPUE hotspots showed significant changes in longitude, with a tendency to continuously expand to the East. The study considers that this phenomenon is mainly due to the influence of the warm pool movement caused by the ENSO phenomenon. During La Niña years (2010–2012), the CPUE hotspots of skipjack were concentrated in the western sea area of 151° E–163° E (west of the boundary line shown in Figure 4), and then, under the influence of El Niño years (2014–2016), the hotspots gradually expanded to the east (Figure 6).

During the change process, the CPUE hotspots in the western part of the study area have always existed, forming a stable fishing ground, and showing consecutive hot spot patterns and persistent hot spot patterns (west of the boundary line shown in Figure 4). The study finds that the Eastern Pacific was gradually transformed from a cold spot to a hot spot due to the warm pool movement, and the significant effect of the eastern position of the hotspot region that has already been shown as a hot spot is continuing to increase and spread continuously to the East, gradually forming a stable fishing ground in the central area of the study area. This region has historically exhibited cold spots, hence its oscillating hot spot pattern in emerging hot spot analysis (east of the boundary line shown in Figure 4). Skipjack, as a thermoregulating fish species, has a fishing ground that changes longitudinally with the variation of the 29 °C isotherm line, and the ENSO phenomenon can affect the position of the 29 °C isotherm line on the edge of the warm pool [11,13], which in turn affects the catch hot spot position of skipjack. Numerous studies [10,12,40,41] have shown that the center of gravity of Pacific skipjack catch is biased towards the West and North during La Niña years, and relatively biased towards the East and South during El Niño years, which is consistent with the findings of this study.

From an annual time scale perspective, the pattern changes around the hot spot area are quite complex. The significant changes in cold or hot spots in this region do not follow a uniform pattern and cannot be classified into a given pattern type, thus they are classified as no pattern detected. These undetected patterns are distributed relatively evenly around the hotspot area, with a distance of approximately 1.5°–3° from the stable hotspot fishing grounds (oscillating hotspots, consecutive hotspots, and persistent hotspots). The sea area near the boundary is inhabited by a substantial population of plankton and zooplankton, thanks to phenomena such as upwelling and subsidence currents. This creates favorable conditions for the aggregation and growth of bonito. The occurrence of "No Pattern Detected" features at this location (see Figure 4, regions I-IV) is likely influenced by the abundance of bait, with food availability driving the distributions. Under normal conditions or during La Niña events, there is a cold-water tongue with high chlorophyll content in the eastern Pacific Ocean, which contains abundant microorganisms and nutrients. This area tends to form good fishing grounds near 160° E. During El Niño events, the fishing grounds move eastward and swing around 165° E [10]. The movement of the warm pool in the western Pacific and the strength of the cold tongue in the eastern Pacific cause the fishing grounds to continually swing and complex variations to occur around the stable fishing grounds. This study's results indicate that the fishing grounds move around the central area due to climate influence, with an oscillation amplitude of 3°–6° in the longitude direction and 1°–2° in the latitude direction (see Figure 4, regions I-IV).

### 4.2. The Impact of Fishing Behavior and Management on Temporal and Spatial Patterns of Fishing Grounds

When decomposing the movement path of the skipjack CPUE hotspot year by year, it can be seen that the skipjack CPUE hotspot as a whole has maintained an eastward movement over the decade (Figure 6). The eastward movement of the fishery is not only

influenced by climate but also disturbed by anthropogenic factors such as fisheries management policies but the proportion of the impact caused by the two factors cannot be quantified for the time being. The results of the hotspot analysis in this study show (Figure 4) that there are consecutive hotspots and intensifying hotspots in the study area, which represent a continuous and stable fishery. However, there is an annular "No Pattern Detect" around the stable fishery (see Figure 4, regions I-IV). Continuous fishing grounds usually occur in a continuous and similar environmental context, but the appearance of the "No Pattern Detect" phenomenon is a break from the norm. The timing, extent, and magnitude of the "No Pattern Detect" phenomenon are highly consistent with the movement of the fishery during the ENSO event, so we believe that the left-right oscillation of the fishery in the study area over the decade was mainly influenced by climatic factors such as ENSO. However, upon closer examination through annual decomposition, we posit that the continuous eastward movement of skipjack CPUE hotspots is primarily attributed to anthropogenic factors.

In the time span covered by this study, there was an increase in the number of Chinese fishing vessels between 2010 and 2013. After 2014, it tended to stabilize under the influence of policies. In 2019, the number of purse seine fishing vessels and fishing days for Pacific skipjack decreased due to objective factors. However, in this study, the expansion trend of fishing hotspots for Pacific skipjack has continued eastward over time (Figure 6), and the fishing days have not had a significant impact on this trend (Table 1). Spatially, the fishing hotspots for Pacific skipjack are highly concentrated, with minimal scattered hotspots. Most of the significant cold spots are located in the boundary areas of the study area. Among the cold spot patterns, the highest proportion is the intensifying cold spot patterns, followed by persistent cold spot patterns and consecutive cold spot patterns (Table 4). These cold spots are relatively evenly distributed around the central fishing grounds depicted in Figure 4. Fishery research inherently possesses strong spatial characteristics, as fishing activities target specific resources in specific areas. In these suitable areas, the catch can reach its maximum [42,43], indicating that the fishing patterns of Chinese fishing vessels are relatively mature.

Hampton [44] used the release and recapture of tagged Pacific skipjack in the pole-and-line fishery to confirm that the population of Pacific skipjack migrates eastward on a large scale during El Niño years and migrates in the opposite direction during La Niña years. Comparing the fishing patterns of Chinese fishing fleets during the study period, it is observed that during La Niña years (2011–2013), the fishing grounds were predominantly located in the western region of the central and western Pacific, including the exclusive economic zones of countries like Papua New Guinea, Micronesia, and Nauru. However, during the El Niño phenomenon from 2014 to 2016, the fishing grounds shifted to the eastern region, such as the exclusive economic zones of Nauru and Kiribati (Figure 6). The purse seine fishing grounds for skipjack in the central and western Pacific Ocean are primarily within the tropical waters between 140° E–150° W and 10° N–10° S, encompassing the exclusive economic zones of eight Pacific island countries, including Papua New Guinea, Micronesia, and Kiribati. The catch of Pacific skipjack in the exclusive economic zones of the eight countries accounts for more than 98% of the total catch in the Pacific island countries [45].

The Nauru Agreement, signed in 1982 by eight Pacific island countries, including Papua New Guinea, Micronesia, and Kiribati, led to the implementation of the "Vessel Day Scheme" (VDS) by the Parties to Nauru Agreement (PNA) in 2007. The VDS system is designed to manage the purse seine fishery [46]. The Chinese tuna purse seine fleet has been developing since 2001. In recent years, with the advancement of the VDS system and other management measures, the price of fishing days in PNA island countries has been increasing. Fishing companies must determine the quantity of fishing days to purchase for each country for the following year by the end of the preceding year. The study period from 2011 to 2016 covers a complete ENSO cycle. The longitude of the fishing grounds has

fluctuated significantly due to the effects of climate change, as shown in the spatiotemporal pattern analysis, with the hotspot shifting eastward (Figure 6). This indicates that the Chinese fleet has gained a preliminary understanding of the general migration pattern of Western and Central Pacific tuna in response to climate change, and can adjust its fishing activities accordingly. In the three years after the end of the ENSO event (2017–2019), the fishing hotspots continued to expand eastward. This trend is likely influenced by the VDS system, as Chinese fishing companies continue to explore new fishing grounds to the East.

## 5. Conclusions

This study explores the spatiotemporal evolution patterns of fish CPUE in the central and western Pacific skipjack tuna purse seine fishery using the spatiotemporal cube model and emerging hotspot analysis method for the first time. It analyzes the hot and cold spot types of fish CPUE in different years and evaluates their trends, reflecting the dynamic spatial and temporal changes of fish CPUE more intuitively. The results obtained from this study contribute to existing research by complementing and verifying previous findings. They serve as a reference for the development of the central and western Pacific tuna purse seine fishery.

The spatiotemporal cube model and emerging hotspot analysis tool used in this study provide several advantages over traditional one-dimensional fishery distribution maps [9]. These include (1) Intuitive visualization: The spatiotemporal cube model and emerging hotspot analysis tool offer a dynamic and visual representation of the time, location, and trend of changes in the target resource. This allows for a more comprehensive understanding of when and where the research objective undergoes specific changes. (2) Comprehensive analysis: Traditional methods often identify a single type of hot or cold spot, providing limited insights into the complexity of spatiotemporal patterns. In contrast, the tool used in this study can identify multiple types of hot and cold spots that change over time, offering 17 pattern classifications. This enables a more detailed and nuanced analysis of the areas where fish CPUE increases or decreases. (3) Improved trend evaluation: The spatiotemporal cube model and emerging hotspot analysis method used in this study enable the evaluation of trends in fish CPUE hot and cold spots over different years. This provides valuable insights into the dynamic spatial and temporal changes of fish CPUE, allowing for a more accurate assessment of resource fluctuations.

Furthermore, future research will incorporate additional data, such as unit fishing effort and environmental/climate factors, to further explore the spatiotemporal evolution patterns and underlying reasons for the tuna resource dynamics. It is important to note that while the three-dimensional visualization of the spatiotemporal cube offers improved observation capabilities, the method for selecting the spatial neighborhood distance and time interval during the construction of the spatiotemporal cube requires further exploration in future research. This will help refine and enhance the accuracy and applicability of the tool.

**Author Contributions:** Conceptualization, R.X. and X.Y.; methodology, R.X. and X.Y.; software, R.X.; validation, R.X.; formal analysis, R.X. and X.Y.; investigation, R.X.; resources, X.Y.; data curation, R.X.; writing—original draft preparation, R.X.; writing—review and editing, R.X. and X.Y.; visualization, R.X.; supervision, X.Y.; project administration, X.Y. and S.T.; funding acquisition, X.Y. and S.T. All authors have read and agreed to the published version of the manuscript.

**Funding:** This research was funded by National Key Research and Development Program of China (2020YFD0901202, 2019YFD0901502).

**Data Availability Statement:** The data that support the findings of this study are available from the corresponding author upon reasonable request.

**Acknowledgments:** This project was funded in part by National Key Research and Development Program of China. We thank all our colleagues from the Research Laboratory for their work in data collection.

**Conflicts of Interest:** The authors declare no conflict of interest.

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
