# Peer review of "Use of Space-Time Cube Model and Spatiotemporal Hot Spot Analyses in Fisheries—A Case Study of Tuna Purse Seine"

_fishes, doi:10.3390/fishes8100525_

Round 1

Reviewer 1 Report

The topic of the manuscript is relevant and has important practical significance. A large amount of factual data was analyzed using adequate modern methods. The reliability of the results obtained is beyond doubt. The manuscript can be recommended for publication in the journal Fishes without any modification.

The only note: Figure 3 is very fuzzy and hard to read. The same applies to Figures 4 and 5, but they need to be enlarged, then perhaps the legends will become clear. And in the list of references, DOIs are not given for all articles.

As a wish, I recommend that the authors, in the Conclusion section, briefly and clearly formulate a list of the main conclusions of the article.

Author Response

For research article

Response to Reviewer 1 Comments

1. Summary

We greatly appreciate your valuable time and effort in reviewing our manuscript. Your feedback and suggestions have been highly valuable and immensely helpful. We have carefully reviewed your comments and have made the necessary revisions accordingly. Please refer to the detailed responses below, as well as the highlighted revisions in the resubmitted document.

2. Questions for General Evaluation

Reviewer’s Evaluation

Response and Revisions

Does the introduction provide sufficient background and include all relevant references?

Yes

Are all the cited references relevant to the research?

Yes

Is the research design appropriate?

Yes

Are the methods adequately described?

Yes

Are the results clearly presented?

Yes

Are the conclusions supported by the results?

Yes

3. Point-by-point response to Comments and Suggestions for Authors

Comments 1: The only note: Figure 3 is very fuzzy and hard to read. The same applies to Figures 4 and 5, but they need to be enlarged, then perhaps the legends will become clear.

Response 1: Thank you for pointing this out. We agree with this comment. We have made revisions to Figure 3, Figure 4, and Figure 5, increasing their resolution to ensure clear visibility of the content. The original figures have been attached for your reference. The revised manuscript will be sent to the journal editor.

Comments 2: And in the list of references, DOIs are not given for all articles.

Response 2: Thank you for pointing this out. We have supplemented the missing DOIs in the references.

Comments 3: As a wish, I recommend that the authors, in the Conclusion section, briefly and clearly formulate a list of the main conclusions of the article.

Response 3: Thank you very much for your valuable comment. We also recognize the importance of this point. In view of the article's length constraints, we have made adjustments to certain paragraph structures to enhance this section.

"This study explores the spatiotemporal evolution patterns of fish CPUE in the central and western Pacific skipjack tuna purse seine fishery using the spatiotemporal cube model and emerging hotspot analysis method for the first time. It analyzes the hot and cold spot types of fish CPUE in different years and evaluates their trends, reflecting the dynamic spatial and temporal changes of fish CPUE more intuitively. The results obtained from this study contribute to existing research by complementing and verifying previous findings. They serve as a reference for the development of the central and western Pacific tuna purse seine fishery.

The spatiotemporal cube model and emerging hotspot analysis tool used in this study provide several advantages over traditional one-dimensional fishery distribution maps [9]. These include: (1) Intuitive visualization: The spatiotemporal cube model and emerging hotspot analysis tool offer a dynamic and visual representation of the time, location, and trend of changes in the target resource. This allows for a more comprehensive understanding of when and where the research objective undergoes specific changes. (2) Comprehensive analysis: Traditional methods often identify a single type of hot or cold spot, providing limited insights into the complexity of spatiotemporal patterns. In contrast, the tool used in this study can identify multiple types of hot and cold spots that change over time, offering 17 pattern classifications. This enables a more detailed and nuanced analysis of the areas where fish CPUE increases or decreases. (3) Improved trend evaluation: The spatiotemporal cube model and emerging hotspot analysis method used in this study enable the evaluation of trends in fish CPUE hot and cold spots over different years. This provides valuable insights into the dynamic spatial and temporal changes of fish CPUE, allowing for a more accurate assessment of resource fluctuations.

Furthermore, future research will incorporate additional data, such as unit fishing effort and environmental/climate factors, to further explore the spatiotemporal evolution patterns and underlying reasons for the tuna resource dynamics. It is important to note that while the three-dimensional visualization of the spatiotemporal cube offers improved observation capabilities, the method for selecting the spatial neighborhood distance and time interval during the construction of the spatiotemporal cube requires further exploration in future research. This will help refine and enhance the accuracy and applicability of the tool. "

4. Response to Comments on the Quality of English Language

Point 1: I am not qualified to assess the quality of English in this paper

Response 1:

5. Additional clarifications

Reviewer 2 Report

The authors present an interesting work about the distribution of hotspots for skipjack tuna in the Central and Western Pacific. In particular, they investigate the behaviour of Chinese fleet during years from 2010 to 2019 applying a space-time cube model, which reveals 13 spatiotemporal patterns in skipjack CPUE. The work is of interest and develop an innovative methodology. I think the work deserves for publication, however, from my point of view, some adjustments are needed.

Abstract

I think this section is complete. I don’t have any comments to add.

Introduction

I think this section can be improved. First, it is important to explicitly define what you intend for hotspots and coldspots. Then, I suggest adding more details about the skipjack species and the effects that ENSO can have on this species. In the pdf files, you can see my suggestion to move part of the discussion to the introduction section.

Materials and methods

I have found this section quite detailed. Some comments are included in the pdf.

Results

Also this section is clear for me. I suggest creating a bullet points for lines from 279 to 330.

Discussion

I think this section can be improved. Specifically, I think a better explanation is needed for the relationships between CPUE and ENSO.

Please have a look of the attached pdf, detailed comments are included in the document.

From my point of view, English is satisfactory; minor editing is required.

Author Response

For research article

Response to Reviewer 2 Comments

1. Summary

On behalf of all the contributing authors, I would like to express our sincere appreciations of your letter and reviewers’ constructive comments concerning our article entitled “Spatiotemporal distribution patterns of skipjack tuna (Katsuwonus pelamis) in the The Central and Western Pacific” (Manuscript No: fishes-2625470). These comments are all valuable and helpful for improving our article. We have carefully reviewed each of the reviewer's comments, and we would like to provide the following explanations regarding the modifications and additions made.

2. Questions for General Evaluation

Reviewer’s Evaluation

Response and Revisions

Does the introduction provide sufficient background and include all relevant references?

Can be improved

We have added this section based on the reviewer's suggestion, please refer to the response 1 for more information.

Are all the cited references relevant to the research?

Yes

Is the research design appropriate?

Yes

Are the methods adequately described?

Yes

Are the results clearly presented?

Yes

Are the conclusions supported by the results?

Can be improved

Thanks for your valuable suggestions, and we have rewritten parts of chapters 4.1 and 4.2 based on the reviewers' suggestions (see line 391-402, 463-472 of the manuscript).

3. Point-by-point response to Comments and Suggestions for Authors

Comments 1: Introduction: I think this section can be improved. First, it is important to explicitly define what you intend for hotspots and coldspots. Then, I suggest adding more details about the skipjack species and the effects that ENSO can have on this species. In the pdf files, you can see my suggestion to move part of the discussion to the introduction section.

Response 1: We think this is a good suggestion. We have added to this section the unique significance of refined cold spots and hot spots (see lines 92-94), the principle of action for ENSO to have an effect on skipjack tuna (see lines 66-72), and moved what you mentioned in the PDF to the Introduction section.

Comments 2: Materials and methods: I have found this section quite detailed. Some comments are included in the pdf.

Response 2: Thank you for your careful review and suggestions, which we have revised or answered based on the notes in the pdf, which can be found in the attached resubmitted manuscript.

Comments 3: Results: Also this section is clear for me. I suggest creating a bullet points for lines from 279 to 330.

Response 3: We sincerely appreciate your suggestion. Symbol lists have been created and displayed in separate segments at this location. (see lines 302-337)

Comments 4:

1. Discussion: I think this section can be improved. Specifically, I think a better explanation is needed for the relationships between CPUE and ENSO.

2. lines 370-372 in the pdf : “This study highlights the regularity observed in the distribution of these spots, indicating a close relationship between small-scale climate changes and the dynamic shifts in catch.”

To me it is not clear the relationship between small-scale changes and the dynamic shifts in catch.

Response 4:
We believe that the two comments are consistent, so we have put them together to answer the question. We sincerely thank you for pointing out this problem. In our perspective, the term "small-scale" here refers to “interannual scale”, and we apologize for the confusion caused by the misrepresentation, and we have made the necessary revisions in the manuscript. (lines 404 in the new manuscript)

We believe that climate change induced by ENSO events at the interannual scale are closely related to the dynamic fluctuations of skipjack CPUE.

In our study, using space-time cube  and spatiotemporal hotspot analysis techniques, we have identified the presence of "No Pattern Detect" areas surrounding both oscillating hotspots and persistent hotspots, as illustrated in Figure 4 regions I-IV.

From the perspective of the formation patterns of fishing grounds, it is evident that the environmental conditions within small-scale marine areas exhibit continuity. Drawing upon established empirical knowledge, such as the Tobler's First Law [1], it can be inferred that under continuous and similar environmental backgrounds, the occurrence of continuous fishing grounds is highly probable.

However, within the findings of this study, the "No Pattern Detect" phenomenon emerges around the stable fishing grounds area. This pattern represents the alternating appearance of fishing cold and hotspots in the given time intervals within the region. The frequency is not fixed, and there is no consistent regularity. Additionally, the scope and magnitude of this phenomenon are consistent with the characteristics of fishing ground movement in terms of amplitude and range during ENSO events. Consequently, we attribute this phenomenon to the spatial response of CPUE to ENSO events.

This study attempts to analyse the relationship between ENSO and CPUE by constructing a spatio-temporal cube and spatio-temporal hotspot analysis technique, which is capable of obtaining the dynamics of a certain spatial location and the different spatial patterns appearing at that location over time (e.g., Fig. 4), but it is difficult to accurately quantify and analyse the relationship for the present purpose.

Thank you again for your valuable suggestions, and we have rewritten parts of chapters 4.1 and 4.2 based on the reviewers' suggestions (see line 391-402, 463-472 of the manuscript).

[1]Tobler, W.R. A Computer Movie Simulating Urban Growth in the Detroit Region. Econ. Geogr. 1970, 46, 234-240, doi:10.2307/143141.

Comments 5:

lines 50-54 in the pdf :”For instance, scholars use the Generalized Linear Mixed Model (GLMM) to calculate the abundance hot-spot index of the target species [4], study the spatiotemporal distribution pattern of the target species by constructing Standard Deviational Ellipses (SDE) [5], or identify fishing hotspots by studying fleet patterns [6] “

I suggest to modify the text as follows: “At present, Generalized Linear Model (GLMM) is one of the most common methodology to calculate the abundance hot-spot index of the target species [4], TO study ....., or TO identify ... patterns [6] “

Response 5:

We have re-written this part according to the Reviewer’s suggestion.

lines 50-54 in the new manuscript :”At present, the Generalized Linear Model (GLMM) is one of the most common methodologies used to calculate the abundance hot-spot index of the target species [4]. It is also relatively common to study the spatial and temporal distribution patterns of the target species by constructing Standard Deviation Ellipses (SDEs) [5], or to identify fishing hotspots by studying fleet patterns [6].”

Comments 6:

lines 60 in the pdf :”other fishing fields”

Not clear to me. Maybe 'for other species' or 'other fishing gear'?!

Response 6:

This means here is 'other fishery fields'. Currently, we have found only a limited number of similar studies in the trawl fishery field, while in other fishing methods such as purse seine and longline fishing, we have not yet come across similar research. We have revised the wording accordingly, and we appreciate your question.

Comments 7:

lines 70-73 in the pdf :By using the space-time cube model and spatiotemporal hot spot analysis technology in GIS, not only can the hot spots or cold spots of the target attribute value be found, but also the intensity and consistency of the hot spots or cold spots within a certain time unit can be determined.

Please rephrase this sentence.

Response 7:

We have re-written this part according to the Reviewer’s suggestion.

lines 85-91 in the new manuscript :”By using the space-time cube model and the spatiotemporal hot spot analysis technique in GIS, we can not only find the hot or cold spots of the target attribute values, but also determine the intensity and consistency of the hot or cold spots within a certain time unit in order to identify the statistically significant hot and cold spot trends over time. By analyzing the data, the tool can identify new, consecutive, intensifying, persistent, diminishing, sporadic, oscillating, or historical patterns of hot or cold spots in different time intervals.”

Comments 8

lines 106-108 in the pdf : “The number of fishing vessels includes leased fishing vessels, bilateral agreements fishing vessels, and “one-card” fishing vessels permitted under the Micronesia Arrangement. “

Sorry, maybe I don't know these specific features of your study area, but the names of the different fishing gear are a bit strange to me. Also, previously you have stated that for this study you have taken into account only purse seines.

Response 8:

Thank you for raising this question. The meaning expressed herein is that the Individua.leets presented herein cover vessels that are flagged to that nation and those vessels considered tobe 'chartered' according to WCPFC Conservation and Management Measures (CMM-2019-08on Charter Notification). The terms in question have been removed, as they merely represent several methods of actual fisheries leasing. These methods do not impact the results of our study and can be disregarded.

Comments 9:

lines 111 in the pdf :” This study conducted preprocessing on the target data,”

Rephrase this sentence

Response 9:

We have re-written this part according to the Reviewer’s suggestion.

lines 129-132 in the new manuscript :“ We initiated our study with data preparation and preprocessing steps, where we adjusted the spatial resolution of the target dataset to 0.5° * 0.5°. The data had a daily time resolution, and we integrated the catch production and fishing effort data on a daily basis within the 0.5° * 0.5° spatial cells.”

Comments 10:

lines 116 in the pdf :“ In the equation: 'Ucatch' represents the cumulative catch within a unit”

What do you mean for 'a unit'? A number of vessels?!

Response 10:

Thank you very much for pointing out this detail. In this context, “unit” represents a spatial cell of 0.5° * 0.5°. We have decided to replace “unit” with “spatial cell of 0.5° * 0.5°” to ensure accuracy and clarity in the content.

Comments 11:

lines 151 in the pdf:“ with positive values indicating growth”

Maybe 'higher presence'?!

Response 11:

Yes, we agree with that expression, it has been modified here.

Comments 12:

lines 163-164 in the pdf:“ The Getis-Ord Gi* statistic is a novel tool used for spatiotemporal hotspot analysis of data in the space-time cube. “

Is it possible to include a reference for the Getis-Ord Gi approach or more details?

Response 12:

As suggested by the reviewer, we have added more references to support this idea ([31-33]).

Comments 13:

lines 201-202 in the pdf:“ This distance was selected as the optimal neighborhood distance for constructing the space-time cube.”

Is there a criteria to how select the optimal neighborhood distance?

Response 13:

In fact, there isn't a fixed standard for selecting the neighborhood distance. Typically, multiples of spatial grid cells are chosen. Through our extensive experimentation, taking into consideration the calculated p-values and z-scores, we found that setting the neighborhood distance to 55500 meters yielded the best results. Due to space limitations, the experimental process was not presented.

Comments 14:

lines 216-217 in the pdf: “ The space-time cube aggregated the 28,490 CPUE data points into 5,310 fishnet grid locations over 10 time step intervals. Each location represented a square area of 55,500 meters by 55,500 meters.”

Maybe 'fishing groundì?!  'Fishing area'?!

Response 14:

In the construction process of the space cube, you need to choose to aggregate the input element point data into the surface grid, this study chose the fishing net grid, i.e., the input elements are to be aggregated into a square (fishing net) polygon with a size of 0.5*0.5. Thank you for pointing out this problem, considering the accuracy of the article's expression, we will change here to “0.5° * 0.5° spatial cells”.

Comments 15:

lines 226-229 in the pdf: “ The 2D visualization provided an overview of the overall trend throughout the study period, while the 3D visualization displayed the historical state and changes of each fishing net location over time. “

I don't think 'fishing net location' is appropriate. I suggest 'fishing ground' or 'fishing area'

Response 15:

Thank you for your suggestion; indeed, “net” is not suitable in this context. We have decided to modify this to “fishing ground represented by each 0.5° × 0.5° spatial cell.”

Comments 16:

lines 238-239 in the pdf: “ a total 238 of 568 fishing net locations exhibit a significant upward trend in CPUE, “

See the previous comments. I don't think “net” is appropriated

Response 16:

Thank you for pointing out this problem, considering the accuracy of the article's expression, we will change here to “spatial cells of 0.5° * 0.5°”.

Comments 17:

lines 329-330 in the pdf: “The occurrence of oscillating hotspots indicates that the frequency and amount of fishing catches in this area are irregular and mainly influenced by environmental changes. “

Can you add a reference for this?

Response 17:

Thank you for your careful checks, here are the results of our research findings, I have reworded this sentence due to a misunderstanding caused by my expression error.

lines 352-354 in the new manuscript:” In this study, we believe that the occurrence of oscillating hotspots indicates that the frequency and amount of fishing catches in this area are irregular and mainly influenced by environmental changes.”

Comments 18:

lines 353-354 in the pdf: “ This study investigates the temporal and spatial dynamics of the CPUE of Western and Central Pacific skipjack”

Here I think we need to include that CPUE refers only to the Chinese vessels.

Response 18:

We think this is an excellent suggestion. We have indicated the “Chinese fishing vessel operations “at the end of this sentence.

Comments 19:

lines 365-367 in the pdf: “ The survival activities of skipjack are closely related to climate changes, and we hypothesize that this oscillatory phenomenon is primarily influenced by the ENSO phenomenon. “

Add a reference for this sentence.

Response 19:

As suggested by the reviewer, we have added more references to support this idea ([35-37]).

Comments 20:

Lines 436-444 in the pdf: “The development of the Pacific skipjack fishery in the central and western Pacific Ocean has seen continuous growth since the 1950s. This can be attributed to advancements in fishing gear and methods, increased fishing vessels, and more fishing days [32]. From 2010 to 2013, there was an increase in the number of fishing vessels, which stabilized after 2014. In line with the requirements of international regional fisheries organizations for sustainable fisheries, China has implemented measures to control the scale of its Pacific skipjack fishery since 2016, resulting in stable numbers of fishing vessels and production capacity [33]. In 2019, the number of purse seine fishing vessels and fishing days for Pacific skipjack decreased due to objective factors.”

Response 20:

We think this to be a valuable suggestion. We have already incorporated most of the content of this paragraph into the introduction section (lines 76-83) and have briefly summarized it here to introduce the following discussion (lines 475-477).

4. Response to Comments on the Quality of English Language

Point 1: From my point of view, English is satisfactory; minor editing is required.

Response 1: Thanks for your suggestion. We have revised the manuscript for grammatical and spelling errors and tried our best to polish the language in the revised manuscript.

5. Additional clarifications

We are very grateful to the reviewers for their careful reading and valuable suggestions. And we apologize for our carelessness. The capitalization issue has been corrected in our resubmitted manuscript. Thanks for your correction. The rewriting and phrase substitution issues marked in the PDF have all been revised and are marked in the manuscript.

Reviewer 3 Report

General comments:

The authors decided to try to apply, for the first time, space-time cube model and spatiotemporal hot spot analyses, as a tool, on official purse seine fishery related data from database available to them. They decided to use data from fishery targeting skipjack to construct data matrix in order to investigate the applicability of the space-time cube model.

Collection of data used has not been previously designed specifically for this study, and therefore the accuracy of data used might be questionable for several reasons (i.e. it is not clear how catch data available in tail number were converted in catch biomass (tons) and also there is no definition of fishing day). This two information are crucial for accuracy of CPUE data (see equation 1), eventually used in space-time cube model data matrix.

Also, it is not clear how the last part of CPUE data time series (2017-2019) has been impacted by introduction of Vessel Day Scheme (VDS) management regime (as mentioned in lines 485-486). There are also many other factors that introduce additional uncertainty in CPUE data, such as those mentioned in lines 42-44. From the paper is not clear if available catch data from database are related to use od FADs or not? Or data represent an unknown mix of catches with and without FADs? CPUE data are highly affected by difference in fishing strategy: with FADs or without FADs.

Therefore, the basic CPUE data, used in space-time cube model, seems to suffer large uncertainties.

Among the objectives of this paper (lines 84-85) authors mentioned relation between hot spots dynamics and ENSO (environmental variable), but nowhere in the paper ENSO index were compared with spatial distribution of skipjack hot spots locations/dynamics!?

Therefore, if I consider this paper as a paper about skipjack tuna, I would not recommend it for publishing.

However, if this paper is considered from methodological point of view, I can see a very interesting attempt (case study) to use new methods/tools for the first time on fisheries related data set, that merit to be considered for publishing.

So, I would suggest to authors to make a major revision of this interesting paper by highlighting the space-time cube model (as a main subject) and spatiotemporal hot spot analyses, as a potentially very useful tool to be used with fisheries related data also (beside previous used of this tool in other fields such as geography, medicine, public transportation…). In that sense the title of the paper might be changed such as: “Use of space-time cube model and spatiotemporal hot spot analyses in fisheries – skipjack tuna case study” or similar.

Specific comments:

Please check use of reference No.3 mentioned in line 44; this reference is about biology

Text in lines 191-202 seems to be more appropriate for methodology section

Temporal dimension is hidden in Figure 2 in most of cubes – not clear for readers; Figure 6 is nice - present time dimension in much more clear way for readers

Text in lines 367-370, 382-385 and 431-433 should be based on some results, but appropriate analyses are missing in this paper

Author Response

For research article

Response to Reviewer 3 Comments

1. Summary

On behalf of all the contributing authors, I would like to express our sincere appreciations of your letter and reviewers’ constructive comments concerning our article entitled “Spatiotemporal distribution patterns of skipjack tuna (Katsuwonus pelamis) in the The Central and Western Pacific” (Manuscript No: fishes-2625470). These comments are all valuable and helpful for improving our article. We have carefully reviewed each of the reviewer's comments, and we would like to provide the following explanations regarding the modifications and additions made.

2. Questions for General Evaluation

Reviewer’s Evaluation

Response and Revisions

Does the introduction provide sufficient background and include all relevant references?

Can be improved

Relevant additions have been made, please refer to lines 66-72, 76-83, 85-94 of the manuscript for details.

Are all the cited references relevant to the research?

Yes

Is the research design appropriate?

Not applicable

For responses to this section, please see response 1 in the point-by-point response letter.

Are the methods adequately described?

Can be improved

Relevant additions have been made, please refer to lines 129-132, 136-137, 184 of the manuscript for details.

Are the results clearly presented?

Can be improved

Relevant additions have been made, please refer to lines 391-402, 463-472 of the manuscript for details. And we changed Figure 4 to add some information.

Are the conclusions supported by the results?

Must be improved

For responses to this section, please see response 3 in the point-by-point response letter.

3. Point-by-point response to Comments and Suggestions for Authors

Comments 1: Is the research design appropriate? Not applicable

Response 1:

Firstly, we would like to express our gratitude to the reviewing expert for taking the time to evaluate our manuscript and provide valuable suggestions for revision. Your valuable comments played a vital role in revising my manuscript.

Below I would like to briefly describe the framework and purpose of my study to show that the research design is appropriate. First of all, we use real fishery production data with a resolution of days, which is compiled and provided by a professional organization, and many researches use this kind of data, the professionalism and accuracy are guaranteed. Please refer to Response 2 for details of the data.

Our research aims to introduce space-time cubes and spatiotemporal hotspot analysis techniques from the realm of GIS   to detect the cold and hotspot types of skipjack cpue and their trends of change, and to analyze the spatio-temporal evolution of skipjack resources. We aspire to leverage this technique to find out the movement trend of fishing hotspots and hotspot concentration areas to maximize the potential of fishery, and provide reference for people to minimize the cost and shift the focus to more productive areas.

In fishery research, the spatial data in question have a wide latitude and longitude, and it is difficult to analyze them in depth, especially the spatio-temporal dynamics research involving a large range and multiple factors. The characteristics of fisheries data are well matched with spatio-temporal analysis tools in GIS, but such methods are not yet widely used.

Through our analysis, we have observed a distinct eastward shift in hotspots in the Central and Western Pacific. Furthermore, employing spatiotemporal hotspot analysis techniques, we have uncovered a phenomenon known as "No Pattern Detect." We have found that this complex oscillation region aligns with the amplitude and range characteristics of fishing grounds movements during El Niño/Southern Oscillation events. We posit that this occurrence represents the spatial response of CPUE to ENSO events, and this conclusion aligns with existing research. (Unfortunately, due to my lack of ability to express myself, I did not state this conclusion clearly in the Discussion section, and I have rewritten this section according to the reviewer's comments. For a more detailed explanation, please refer to line 391-402, 463-472 in our revised manuscript and my subsequent response in Response 3).
However, there is indeed the problem of difficulty in quantifying the relationship between the two, ENSO and cpue, and thank you for mentioning the idea of comparing the ENSO index with the hotspot location, which is a challenge we are actively addressing.

Comments 2:  

The authors decided to try to apply, for the first time, space-time cube model and spatiotemporal hot spot analyses, as a tool, on official purse seine fishery related data from database available to them. They decided to use data from fishery targeting skipjack to construct data matrix in order to investigate the applicability of the space-time cube model.

Collection of data used has not been previously designed specifically for this study, and therefore the accuracy of data used might be questionable for several reasons (i.e. it is not clear how catch data available in tail number were converted in catch biomass (tons) and also there is no definition of fishing day). This two information are crucial for accuracy of CPUE data (see equation 1), eventually used in space-time cube model data matrix.

Also, it is not clear how the last part of CPUE data time series (2017-2019) has been impacted by introduction of Vessel Day Scheme (VDS) management regime (as mentioned in lines 485-486). There are also many other factors that introduce additional uncertainty in CPUE data, such as those mentioned in lines 42-44. From the paper is not clear if available catch data from database are related to use od FADs or not? Or data represent an unknown mix of catches with and without FADs? CPUE data are highly affected by difference in fishing strategy: with FADs or without FADs.

Therefore, the basic CPUE data, used in space-time cube model, seems to suffer large uncertainties.

Response 2:

We sincerely appreciate your valuable feedback, and I will now provide explanations for each point:
1. The data used in this study is derived from actual production data of Chinese fishing vessels, which has been provided by professional organizations. Using real data for fisheries research is a common practice, and currently, daily-resolution fishing log data is the most accurate data available to us.
2. The subject of this study is purse seine fisheries, where the biological tonnage of the target species is directly obtained through purse seine fishing, and it is counted by weight, eliminating the issue of conversion.
3. The term 'fishing day' represents 'Days fishing and searching (effort),' and this definition aligns with the field definitions in the dataset of the WCPFC. We have already supplemented this information in the manuscript.(lines 138,table 1)
4. In our study, we analyzed the year-by-year decomposition of hotspot movement trends and found that during the period of 2017-2019, fishing hotspots continued to expand eastward. During this time, the ENSO cycle had ended, and the influence of climate change had diminished compared to earlier periods. However, the eastward expansion trend did not weaken. We believe that this phenomenon is primarily influenced by fisheries management policies such as the Vessel Day Scheme (VDS). These regulations require companies to determine the purchase quantity of fishing days for the next year by the end of the previous year. The previous eastward movement of hotspots caused by the ENSO effect attracted Chinese fishing companies to continually explore fishing grounds to the east. This result is consistent with the actual situation.
5. Thank you for this valuable opinion, this study did not differentiate between the use of Fish Aggregating Devices (FADs) or not, mainly due to the limitation of the data, we will consider the introduction of FADs and other related conditions in subsequent studies.

Comments 3: Among the objectives of this paper (lines 84-85) authors mentioned relation between hot spots dynamics and ENSO (environmental variable), but nowhere in the paper ENSO index were compared with spatial distribution of skipjack hot spots locations/dynamics!?  

Response 3:

This study attempts to analyse the relationship between ENSO and CPUE by constructing a spatio-temporal cube and spatio-temporal hotspot analysis technique, which is capable of obtaining the dynamics of a certain spatial location and the different spatial patterns appearing at that location over time (e.g., Fig. 4), but it is difficult to accurately quantify and analyse the relationship for the present purpose.

In our study, using space-time cube  and spatiotemporal hotspot analysis techniques, we have identified the presence of "No Pattern Detect" areas surrounding both oscillating hotspots and persistent hotspots, as illustrated in Figure 4 regions I-IV. From the perspective of the formation patterns of fishing grounds, it is evident that the environmental conditions within small-scale marine areas exhibit continuity. Drawing upon established empirical knowledge, such as the Tobler's First Law [1], it can be inferred that under continuous and similar environmental backgrounds, the occurrence of continuous fishing grounds is highly probable.

However, within the findings of this study, the "No Pattern Detect" phenomenon emerges around the stable fishing grounds area. This pattern represents the alternating appearance of fishing cold and hotspots in the given time intervals within the region. The frequency is not fixed, and there is no consistent regularity. Additionally, the scope and magnitude of this phenomenon are consistent with the characteristics of fishing ground movement in terms of amplitude and range during ENSO events. Consequently, we attribute this phenomenon to the spatial response of CPUE to ENSO events. We believe that climate change induced by ENSO events at the interannual scale are closely related to the dynamic fluctuations of skipjack CPUE.

[1]Tobler, W.R. A Computer Movie Simulating Urban Growth in the Detroit Region. Econ. Geogr. 1970, 46, 234-240, doi:10.2307/143141.

Comments 4: So, I would suggest to authors to make a major revision of this interesting paper by highlighting the space-time cube model (as a main subject) and spatiotemporal hot spot analyses, as a potentially very useful tool to be used with fisheries related data also (beside previous used of this tool in other fields such as geography, medicine, public transportation…). In that sense the title of the paper might be changed such as: “Use of space-time cube model and spatiotemporal hot spot analyses in fisheries – skipjack tuna case study” or similar.

Response 4:

We think this is an excellent proposal. Allow us to express our personal opinion that, in general, space-time cubes have seldom been used to analyse fisheries data which is unfortunate as the characteristics of fishery data lend themselves well to these analyses. This study applies space time cubes and spatial hotspot analysis to a different type of fisheries data compared to previous research, showing a certain level of originality. The study's primary objective remains the exploration of current fisheries issues using innovative methods, and in certain aspects, it may not fit the category of a purely methodological paper. Therefore, we kindly ask the reviewers to consider the appropriateness of our previous title. If the reviewers and editors still feel that a change is necessary, we are more than willing to make the change "Use of space-time cube model and spatiotemporal hot spot analyses in fisheries - a case study of tuna purse seine".Once again, we sincerely appreciate your positive feedback and valuable suggestions for improving the quality of our manuscript.

Comments 5:

Please check use of reference No.3 mentioned in line 44; this reference is about biology.

Response 5:

Thank you for your thorough review. Upon verification, the reference citation in this section is correct. The referenced article is published in a Chinese journal titled 'JOURNAL OF BIOLPGY' and primarily focuses on the advances in the biology of skipjack under the title 'Review on the biology of skipjack tuna Katsuwonus pelamis. The author is a young scholar, and his research has been widely cited.

Comments 6:

Text in lines 191-202 seems to be more appropriate for methodology section.

Response 6:

Thank you for your suggestion. This section primarily analyzes the spatial autocorrelation characteristics of the data to determine that the skipjack CPUE occurs in non-random clusters. While a brief mention is made of the specific implications represented by the spatial autocorrelation calculation results, it is not presented as an independent paragraph due to its relatively small size, are mainly a technical reference. The specific formulas are described in 2.3.1 Spatial Autocorrelation Analysis. Please refer to that section for further information.

Comments 7:

Temporal dimension is hidden in Figure 2 in most of cubes – not clear for readers; Figure 6 is nice - present time dimension in much more clear way for readers.

Response 7:

Thank you for your suggestion. Figure 2 is mainly used as a graphical introduction to the spacetime cube model from section 3.2, where the layer where each cube is located represents a time dimension. Since this is intended as a conceptual illustration, time slices have not been separated.

Comments 8:

Text in lines 367-370, 382-385 and 431-433 should be based on some results, but appropriate analyses are missing in this paper.

Response 8:

We have re-written this part according to the Reviewer’s suggestion.

Text in lines 367-370 :

lines 391-402 in the new manuscript :From the perspective of the formation patterns of fishing grounds, it is evident that the environmental conditions within small-scale marine areas exhibit continuity. Drawing upon established empirical knowledge, such as the Tobler's First Law [1], it can be inferred that under continuous and similar environmental backgrounds, the occurrence of continuous fishing grounds is highly probable.However, within the findings of this study, the “No Pattern Detect” phenomenon emerges around the stable fishing grounds area (see Figure 4, regions I-IV). The “No Pattern Detect” represents the alternating appearance of fishing cold and hotspots in the given time intervals within the region. The frequency is not fixed, and there is no consistent regularity. The location of the “No Pattern Detect” phenomenon and the amplitude of the oscillations are highly consistent with the left-right movement of warm pools in the western Pacific Ocean during ENSO events.

Text in lines 382-385: In conjunction with the additions in the previous paragraph, we do not consider it necessary to repeat the clarification here and therefore leave it unchanged.

Text in lines 431-433:

lines 463-472 in the new manuscript : The results of the hotspot analysis in this study show (Figure 4) that there are consecutive hotspots and intensifyinghotspots in the study area, which represent a continuous and stable fishery. However, there is an annular "No Pattern Detect" around the stable fishery (see Figure 4, regions I-IV). Continuous fishing ground usually occur in a continuous and similar environmental context, but the appearance of the "No Pattern Detect" phenomenon is a break from the norm. The timing, extent, and magnitude of the "No Pattern Detect" phenomenon are highly consistent with the movement of the fishery during the ENSO event, so we believe that the left-right oscillation of the fishery in the study area over the decade was mainly influenced by climatic factors such as ENSO.

4. Response to Comments on the Quality of English Language

Point 1: I am not qualified to assess the quality of English in this paper

Response 1: 

5. Additional clarifications

We tried our best to improve the manuscript and made some changes marked in revised paper which will not influence the content and framework of the paper. We appreciate for Reviewers’ warm work earnestly, and hope the correction will meet with approval. Once again, thank you very much for your comments and suggestions.

Round 2

Reviewer 3 Report

My principal concern in this manuscript was data issue. I am aware of the fact that authors are not able to improve the accuracy of official fisheries data and eliminate uncertainities related to skipjack CPUE calculation. However, I highly appreciate all efforts did by authors in order to improve this manuscript.

Therefore, I am glad to see that authors are willing to change the title of their manuscript in more appropriate one: "Use of space-time cube model and spatiotemporal hot spot analyses in fisheries - a case study of tuna purse seine".

In my opinion, if title will be changed as suggested by authors, I would reccomend this manuscript for publication.